# Improvement of the Thermal Conductivity and Mechanical Properties of 3D-Printed Polyurethane Composites by Incorporating Hydroxylated Boron Nitride Functional Fillers

**DOI:** 10.3390/ma16010356

**Published:** 2022-12-30

**Authors:** Kai-Han Su, Cherng-Yuh Su, Wei-Ling Shih, Fang-Ting Lee

**Affiliations:** 1Institute of Mechatronic Engineering, National Taipei University of Technology, No. 1, Section 3, Zhongxiao East Road, Taipei 106, Taiwan; 2Institute of Physics, Academia Sinica, No. 128, Section 2, Academia Road, Taipei 11529, Taiwan; 3Additive Manufacturing Center for Mass Customization Production, National Taipei University of Technology, No. 1, Section 3, Zhongxiao East Road, Taipei 106, Taiwan

**Keywords:** TPU, hydroxyl-functionalized boron nitride, thermal interface materials (TIMs), three-dimensional (3D) printing, fused deposition modeling (FDM), thermal conductivity

## Abstract

Recently, the use of fused deposition modeling (FDM) in the three-dimensional (3D) printing of thermal interface materials (TIMs) has garnered increasing attention. Because fillers orient themselves along the direction of the melt flow during printing, this method could effectively enhance the thermal conductivity of existing composite materials. However, the poor compatibility and intensive aggregation of *h*-BN fillers in polymer composites are still detrimental to their practical application in thermally conductive materials. In this study, hydroxyl-functionalized boron nitride (OH-BN) particles were prepared by chemical modification and ultrasonic-assisted liquid-phase exfoliation to explore their impact on the surface compatibility, mechanical properties and the final anisotropic thermal conductivity of thermoplastic polyurethane (TPU) composites fabricated by FDM printing. The results show that the surface-functionalized OH-BN fillers are homogeneously dispersed in the TPU matrix via hydrogen bonding interactions, which improve the interfacial adhesion between the filler and matrix. For the same concentration of loaded filler, the OH-BN/TPU composites exhibit better mechanical properties and thermal conductivities than composites incorporating non-modified *h*-BN. These composites also show higher heat conduction along the stand-vertical direction, while simultaneously exhibiting a low dielectric constant and dielectric loss. This work therefore provides a possible strategy for the fabrication of thermal management polymers using 3D-printing methods.

## 1. Introduction

As new modern electronic devices have become more integrated and miniaturized, effective heat dissipation management has become an urgent challenge to be addressed in order to increase the performance, efficiency, and operational lifetime of these devices [1]. The development of thermal interface materials (TIMs) has therefore received considerable attention in recent years. In general, TIMs are polymer resins blended with a highly thermally conductive filler material, as it combines the advantages of different material systems [2]. The resulting composites should possess high thermal conductivity, a low dielectric constant, and high deformability to ensure full contact between two surfaces, thus ensuring efficient heat transfer [3,4]. Thermoplastic polyurethanes (TPU) are one of the most widely used resins in TIM formulations because of their excellent flexibility, elasticity, adhesion and sustainability [5,6,7].

For the formulation of thermally conductive fillers, carbon-based (such as graphene and carbon nanotubes) and ceramic (such as boron nitride (BN), aluminum oxide (Al_2_O_3_), aluminum nitride (AlN), and silicon carbide (SiC)) nanomaterials have been studied extensively [8,9,10,11]. Although carbon-based materials possess high thermal conductivities, they also possess extremely high electrical conductivities, thus causing device malfunction through uncontrolled leakage currents. Conversely, ceramic particles have excellent insulation properties while maintaining high thermal conductivities, making them suitable for use as TIM filler materials in electronics applications [12,13]. Among these, hexagonal boron nitride (*h*-BN), which possesses a two-dimensional (2D) layered structure analogous to graphite, is an attractive material owing to its intrinsically high thermal conductivity and strong anisotropy (the in-plane and through-plane conductivities are approximately 600 and 2–30 W∙m^−1^∙K^−1^, respectively). *h*-BN also benefits from superior mechanical strength, chemical inertness, electrical insulation properties, moderate cost, and a more stable crystal structure than other ceramic materials [14,15,16]. However, achieving a homogeneous dispersion of *h*-BN in the polymer matrix is difficult because of its chemical inertness and high surface energy, consequently reducing the potential thermal conductivity of the resulting composite.

To solve this issue, surface functionalization and exfoliation strategies for *h*-BN have been employed [17]. Surface functionalization is necessary to enhance the wettability, compatibility, dispersion and interfacial interactions of *h*-BN particles within the polymer matrix [18]. Hydroxylation is the most important functionalization route in this regard. Wang et al. prepared a *h*-BN composite with enhanced thermal conductivity and mechanical strength using cross-linked lignin nanoparticles and hydroxylated *h*-BN [19]. Huang et al. reported the hydroxyl functionalization improves the dispersion and compatibility of BNNs in solvents. The B–OH groups on BNNs can readily form hydrogen bonds with the reactive polymer matrix, which allows for a better synergy between the inorganic fillers and polymers [20]. Compared to pure *h*-BN, hydroxylated *h*-BN (OH-BN) exhibits enhanced and unique physical and chemical properties. In particular, the introduction of hydroxyl groups allows the homogeneous dispersion of *h*-BN materials in several polar solvents, thereby promoting their adhesion to polymer matrices [21,22,23]. In addition, many studies have reported various methods for exfoliating *h*-BN to overcome the van der Waals forces between their stacked layers, thus separating them to achieve a wider dispersion. This results in a larger surface area and interfacial volume for the *h*-BN filler as well as a higher thermal conductivity for the resulting composite material [24]. Previously, Kim et al. reported the use of low pressure chemical vapor deposition (LPCVD) synthesis to obtain large-area monolayer of *h*-BN films; however, this requires ultrahigh vacuum systems, a high-temperature atmosphere and specialized templates [25]. Deepika et al. also reported the use of wet-ball milling as an efficient method to produce single-layer boron nitride nanosheets (BNNS), but the high amount of force employed introduced undesirable structural defects into the hexagonal lattice [26]. In addition, Zhi et al. exfoliated pure *h*-BN into highly-polar DMF using intense ultrasonication, followed by centrifugation. However, the yield of usable material from this approach was low [27].

Recently, the combination of surface functionalization and ultrasound-assisted liquid-phase exfoliation has become an ideal approach for exfoliating *h*-BN. By taking OH-BN as an intermediate, and further exfoliating this using ultrasonic waves in polar solvents, OH-BN nanosheets with a relatively small number of hydroxyl groups at their surfaces can be produced. Notably, this approach combines the advantages of mechanical exfoliation and chemical modification to synergistically enhance the dispersion of *h*-BN fillers in polymer composites in a simple process that is easily scalable. It is well known that during the preparation of *h*-BN TIM composites, the orientation of the fillers is one of the key factors that determines the material’s thermal conductivity. Thus, building interconnecting directional fillers with a thermally conductive skeleton within polymers, parallel to the direction of heat flow, has gained attention as an effective strategy for enhancing the performance of TIMs [28,29]. In recent years, many studies have reported various preparation methods for these structures, including doctor blading, injection molding, hot pressing, vacuuming, shear field and magnetic/electrical field alignment [30,31,32,33,34,35,36]. Although these studies have made considerable progress in terms of improving composite performance, these technologies still face challenges in their implementation, owing to the complexity of fabricating parts with intricate internal structures and personalized shapes for diverse thermal management applications.

Recently, 3D printing has attracted interest as an additive manufacturing technology because of its ability to produce customized profiles and complex structures with short lead times [37]. Among the existing 3D printing methods, fused deposition modeling (FDM) printing is a promising approach for forming highly ordered structures of fillers inside composites, as the filler orientation can be influenced by the shear force exerted by the printer nozzle [38,39]. Across recent articles, Luo et al. reported that BN-filled liquid crystal polymer (LCP) composites are oriented to acquire the preset heat conduction network using FDM 3D printing technique, and the thermal conductivity of the composites reached 1.77 W∙m^−1^∙K^−1^ at a BN content of 20 wt.% [39]. Wang et al. also focused on the orientation of the graphene oxide-boron nitride (rGO-BN) hybrid filler, which could be effectively controlled by FDM printing methods; the results show that the thermal conductivity of the composite specimens prepared under three different printing directions can reach 2.61, 1.14 and 0.63 W∙m^−1^∙K^−1^, respectively [40]. However, in such filaments, when composites rely on simply embedding non-modified *h*-BN particles in thermoplastics by melt compounding, additional complexities are incorporated into the mechanical response, such as poor adhesion between a matrix–filler interface and undesirable pores in the composite filaments [41]. For example, Altay et al. prepared a composite material from BN with synthetic graphite (SG). It was found that the tensile strength of the thermally conductive composite materials formed decreased from approximately 70 MPa to less than 60 MPa at a BN content of 50 wt.% [42]. Another issue is the optimization of the filler fractional concentration, which must be sufficient to favor the establishment of a continuous thermal network structure, while avoiding filler accumulation in the nozzle, clogging, brittle filaments and a reduction in print quality [43,44]. Thus, it remains a challenge to prepar’e high-quality TIMs with a combination of excellent mechanical properties, high processability and high thermal conductivity using this method. However, the state of fillers’ dispersion and interfacial compatibility, activated by chemical bonds forming between the filler and polymer matrix, incorporating functionalized fillers in this fabrication method and optimized for homogenous matrix incorporation, may lead to an improvement in composite quality and performance.

To this end, this study has been conducted to examine the effect of the combination of filler modification and orientation on the performance of a composite TIM. To achieve this, TPU composites filled with OH-BN were successfully prepared and their thermal conduction and mechanical properties were investigated. First, the OH-BN filler was obtained by a typical liquid-phase exfoliation method with the assistance of ultrasonication. Second, the ordered orientation of OH-BN was achieved via FDM 3D printing, resulting in the formation of continuous thermal conductive pathways in the TPU matrix. Finally, the microstructure, cross-sectional morphology, thermal properties, and mechanical properties of the OH-BN/TPU composites were systematically investigated for different degrees of filler loading. These results were then compared with those of unfunctionalized *h*-BN/TPU composites to examine the difference in composite performance. The new strategic synergy claimed in this study is facile, and also overcomes the poor dispersion of *h*-BN in the polymer matrix, along with the stringent requirements of orientation for heat transfer in *h*-BN based fillers.

## 2. Materials and Methods

### 2.1. Materials

*h*-BN powder (3–4 μm particle size) was provided by National Nitride Technologies Co., Ltd. (Taichung, Taiwan). NaOH was obtained from Honeywell International Inc. (Charlotte, NC, USA). Thermoplastic polyurethane pellets (Elastollan S 85A) were provided by BASF Co., Ltd. (Ludwigshafen, Germany). All chemicals used in the experiments were of analytical grade and used without further purification.

### 2.2. Preparation of Hydroxyl-Functionalized h-BN (OH-BN)

The *h*-BN particles were dispersed in a 5 M NaOH solution with magnetic stirring at 120 °C for 24 h to functionalize the *h*-BN particle surface with hydroxyl groups. After this treatment, the particles were filtered several times and rinsed with deionized (DI) water to adjust the pH value from basic to neutral. The OH-BN particles were then heated in a furnace at 80 °C for 5 h to remove any residual water. Finally, the OH-BN particles were dispersed in DI water with ultrasonication for 20 min, followed by centrifugation at 3000× *g* rpm for 15 min. The resulting particles were dried at 60 °C in a vacuum oven for 8 h and stored in a vacuum desiccator before use.

### 2.3. Preparation of the TPU Composite Filaments

The TPU pellets were dried in a vacuum oven for 8 h at 80 °C under a pressure of 700 mmHg to remove absorbed water vapor prior to compounding. Afterward, the pellets were mixed with varying amounts of OH-BN using a twin-screw extruder (Kobelco Co., Ltd., Tokyo, Japan) at a processing temperature of 170 °C and a rotation speed of 140 rpm for 20 min to prepare TPU composites with two different filler weight percentages. The resulting composites were cut into small pellets and further extruded using a single-screw extruder (Geor-Ding Machinery Co., Ltd., Taichung, Taiwan). The extruding temperatures were set to 200, 200, 205, 205, and 205 °C (from feeding section to extruding die). The melt pump pressure was set at 50 bar, the extruder screw rotation was set to 12 rpm, a water bath temperature of 18 °C was maintained and the diameter of the extruded filament was maintained at 1.75 ± 0.05 mm by controlling the speed of the puller roller. The same method was also applied to prepare *h*-BN/TPU samples with different filler-loading concentrations as the control samples for the subsequent experiments. The final filler weight–percent concentrations in the OH-BN/TPU and *h*-BN/TPU composite filaments were 5 and 10 wt.%.

### 2.4. Preparation of FDM 3D Printing of TPU Composites

All test specimens were prepared using a Tenlog TL-D3 Pro Independent Dual Extruder DMP 3D Printer (Tenlog 3D Technology Co., Ltd., Shenzhen, China). The nozzle and bed temperatures were set to 220 °C and 60 °C, respectively. The nozzle diameter was 0.4 mm, the printing speed was set to 20 mm/s, and the layer thickness was set to 0.2 mm. All the processing procedures, from component compounding to 3D printing, are shown in Figure 1. The mechanical testing specimens were fabricated with volumes of 115 × 25 × 6 mm^3^ and 20 × 5 × 2 mm^3^, and the nozzle scanning direction for each layer was parallel to the through-plane direction. Two different printing methods were used for preparing the thermal conductivity testing samples, as schematically shown in Figure 2a,b. These were differentiated by the resulting orientation of the filler network structure inside the composite. The designated flat-plane (FP) and stand-vertical (SV) specimens were printed with a horizontal or vertical network orientation relative to the print bed plane, respectively.

### 2.5. Composite Material Characterization

A Fourier transform infrared (FT-IR) spectrometer (FT/IR-4150, JASCO, Tokyo, Japan) was used to analyze the chemical structures of the unmodified *h*-BN, the functionalized OH-BN filler particles, the thermally conductive TPU composites and the surface-treated materials. The vibrational spectra were acquired between 400 and 4000 cm^−1^ with a spectral resolution of 4 cm^−1^. The microstructures and morphologies of *h*-BN before and after the surface treatment and the dispersion state of the fillers in the TPU matrix were examined using field-emission transmission electron microscopy with an acceleration voltage of 5 kV (FE-SEM, JSM-7610F, JEOL, Tokyo, Japan). Raman spectroscopic analysis was performed using a Fourier spectrometer (ACRON, Confocal Micro RAMAN mapping system, Seoul, Korea) with a single longitudinal mode (SLM) laser (λ = 532 nm, power = 100 mW cm^−2^) to examine the structural changes in OH-BN before and after exfoliation. Based on the transient plane source (TPS) method, a thermal constant analyzer (Hot Disk TPS 2500, Gothenburg, Sweden) was used to determine the thermal conductivity of the composites, and all measuring processes followed the international standard: ISO 22007-2 [6,8,11]. At least three separate measurements were performed for each sample and the average conductivity was calculated. The stress–strain performance of the composite samples was measured using a universal testing machine (Instron4464, Instron Corp., Norwood, MA, USA) at a cross-head speed of 5 mm/min. For each composite, a dumbbell-shaped sample with dimensions of 115 × 25 × 6 mm^3^ was printed for tensile testing according to ASTM D638 international standards. All measurements were carried out at room temperature (25 °C). For each composition, at least five samples were tested and their values averaged. A dynamic mechanical analyzer (DMA Q800, TA Instruments, New Castle, DE, USA) was used to determine the dynamic mechanical properties and glass transition temperature (T_g_) of the composite specimens. The stretch mode temperatures were varied from −100 to 150 °C. The heating rate and frequency were set at 5 °C/min and 1 Hz, respectively. The rectangular samples for these measurements were fabricated with dimensions of 20 × 5 × 2 mm^3^. The dielectric constant and dielectric loss were measured using a precision impedance analyzer (Agilent 8722ES, Agilent Technologies Inc., Santa Clara, CA, USA) at room temperature within a frequency range of 5 MHz to 1 GHz. The dimensions of the flake samples cut from the central part of the dumbbell-shaped specimens were 80 × 80 × 1 mm^3^.

## 3. Results and Discussion

### 3.1. FT-IR

Figure 3a shows the FT-IR spectra of the unmodified *h*-BN and surface-functionalized OH-BN particles. Here, both *h*-BN and OH-BN exhibit two sharp characteristic peaks at approximately 1366 and 801 cm^−1^, corresponding to the B-N in-plane stretching and B-N-B out-of-plane bending vibrations, respectively [45]. Because the OH-BN particles were hydroxylated at the edges of the lattice planes by the NaOH treatment, a broad peak was observed at approximately 3430 cm^−1^ for this material, attributed to the stretching vibration of the B-OH bonds [30,46]. This confirmed that hydroxyls were successfully grafted onto the *h*-BN lattice.

Figure 3b displays the FTIR spectra of pure TPU compared with the *h*-BN/TPU and OH-BN/TPU composites. In the pure TPU spectrum, characteristic peaks were observed at 3430, 3320, 2950, 1735, 1720, 1535, and 1230 cm^−1^. These peaks correspond to the O-H, N-H, and -CH_2_ (saturate carbon) stretching, free and H-bonded carbonyl, C-N asymmetric stretching, and N-H bending absorptions, respectively [28]. The *h*-BN/TPU and OH-BN/TPU composites also displayed similar absorption bands; however, their intensities are stronger owing to the overlapping of the TPU spectrum with the higher-intensity signal of the filler peaks. Here, the characteristic absorption band between 1650–1780 cm^−1^ (displayed in the inset of Figure 3b) can be attributed to the C=O stretching vibration of the carbonyl groups within the TPU urethane linkage. This region also provided information about inter-urethane hydrogen bonding; for the OH-BN/TPU composites, the carbonyl peak intensities were higher than those for the pure TPU and *h*-BN/TPU composites, suggesting the presence of hydrogen bonding interactions between the TPU chains and the OH-BN filler particles [5,47,48,49].

### 3.2. Raman

Raman spectroscopy has been widely used to characterize the interlayer effects of *h*-BN materials, and can provide detailed information about the exfoliation state of the resultant OH-BN particles. Figure 4 presents the Raman spectra of the unmodified *h*-BN and surface-functionalized OH-BN particles. Both spectra exhibited a characteristic peak at approximately 1360 cm^−1^, which can be assigned to the B-N vibrational mode (intralayer E2g phonon mode) [50]. Notably, OH-BN exhibited a small peak shift to approximately 1362 cm^−1^. Exfoliation by sonication in the liquid phase resulted in a further shift to approximately 1365 cm^−1^. This behavior is explained by the weaker interlayer interactions and higher in-plane strain caused by the removal of *h*-BN layers [51,52]. In addition, the peak intensity of OH-BN was substantially weaker and broader than that of the unmodified *h*-BN particles, with the full width at half maximum (FWHM) falling from 23.4 to 15.5 cm^−1^. This result can be attributed to a decrease in the superposition of multiple peaks at approximately 1360 cm^−1^, verifying the considerable reduction in the *h*-BN particle thickness after the exfoliation process [53,54].

### 3.3. FE-SEM

After confirming the surface functionalization and liquid phase exfoliation of the filler particles, we analyzed their surface morphology by using FE-SEM, as shown in Figure 5. The unmodified *h*-BN particles exhibited smooth surfaces with a typical hexagonal plate-like morphology (Figure 5a). Here, it can be seen that the *h*-BN particles were tightly aggregated to form a stacked sheet structure. In contrast, the edges of the functionalized OH-BN particles were rough and retained a typical layered structure, as shown in Figure 5b. These particles also possessed a relatively smaller surface area than the unmodified *h*-BN particles, which may be attributed to the higher degree of OH-BN layer separation during ultrasonication afforded by the appended hydroxyl groups [30,55].

The effect of the surface treatment on the compatibility and dispersion of the filler particles (at a 10 wt.% concentration) within the TPU matrix was investigated by examining the cross-section of the composite filaments, as shown in Figure 5c,d. Here, agglomerated *h*-BN particles were observed on the fracture surface of the *h*-BN/TPU composite filaments (Figure 5c), and many cracks and interfacial gaps were seen at the filler/TPU matrix interface. In contrast, the OH-BN/TPU composite filaments displayed a homogeneous dispersion of filler particles with no apparent interfacial gaps (Figure 5d). This is likely because of the formation of strong hydrogen bonds between the OH-BN hydroxyl groups and the carboxylic groups in the TPU matrix, which would provide stronger interfacial interactions and therefore better interfacial adhesion than those observed in the unmodified *h*-BN/TPU composite [56,57].

We further compared the cross-sectional morphologies of the *h*-BN/TPU and OH-BN/TPU composites, as shown in Figure 5e,f. The SEM images for both composites exhibit that the filler particles were effectively orientated along a single direction, owing to the strong shearing forces imparted by the nozzle during the 3D printing process. However, the *h*-BN fillers were weakly ordered and considerably agglomerated (Figure 5e), which can be attributed to the high surface energy of unmodified *h*-BN. This resulted in poor dispersion and compatibility within the TPU matrix. Conversely, the OH-BN particles presented a more homogeneous dispersion and highly oriented structure in the TPU matrix, forming an efficient continuous filler network (Figure 5f). This also implies that the hydroxyl functionalization of OH-BN provided more continuous heat conduction paths through the TPU matrix, an important factor for improving the thermal conductivity of TPU composites.

### 3.4. Tensile Stress–Strain Testing

Tensile stress–strain tests were performed to investigate the effects of different filler concentrations on the mechanical properties of the FDM 3D-printed constructs. The results are shown in Figure 6. The tensile strength and elongation at break values of the pure TPU specimen were 15 MPa and 950%, respectively. The tensile strengths of all the composite specimens were higher than that of the pure TPU specimen, whereas their elongation at break values decreased with increasing filler concentration. A maximum tensile strength of 22 MPa was reached at a 10 wt.% OH-BN concentration, approximately 1.5 times greater than that of the baseline TPU specimen, while the elongation at break was slightly decreased. This improved tensile stress performance arises from the intrinsically favorable mechanical properties of the fillers, which acted as a skeletal supporting structure in the composite [8]. However, as the OH-BN concentration increases, the filler/matrix interactions restrict the mobility of the TPU chains, resulting in a reduced elongation at break compared with the control specimen [58].

In addition, it was also found that the tensile strength and elongation at break of the OH-BN/TPU composites were notably larger than those of the *h*-BN/TPU composites of equal filler concentration. This suggests that the numerous agglomerations in the latter composites increase the stress concentration at the filler/matrix interface, initiating stress transfer failure in the continuous polymer phase [59,60]. Conversely, hydrogen bonding between OH-BN and the TPU matrix inhibits restacking and aggregation of the filler particles as well as providing greater interfacial adhesion within the matrix [48]. Moreover, the increased contact surface area between the filler and the matrix allows the applied force to be effectively transferred to the OH-BN network, which contributes to the increased mechanical strength in these composites [50].

### 3.5. Dynamic Mechanical Analysis

The dynamic mechanical properties of the OH-BN/TPU and *h*-BN/TPU composites were measured between −50 and 130 °C. Figure 7a shows the storage modulus of the TPU composites according to filler concentration. As expected, all composites exhibited a higher storage modulus than that of the pure TPU matrix. This is because of the high rigidity of the BN-type particles, which confers a general increase in the stiffness of the composite matrix. In each case, the modulus also increased with increasing filler concentration. At equal filler concentrations, the OH-BN/TPU composites exhibited a higher storage modulus than the *h*-BN/TPU composites at the same temperature. This can be attributed to the improved dispersion and interactions of the OH-BN fillers with the TPU polymer chains, which results in a higher and more equally distributed stiffness through the composite matrix. In addition, it was observed that the storage modulus of all the samples decreased with increasing temperature. At low temperatures, polymeric materials are in a glass-like state, where the polymer chains have low mobility and the highest storage modulus value. The mobility of smaller-sized structural molecules increases with rising temperature, resulting in a rapid decrease in storage modulus and a transition into a rubber-like state. In this case, it was observed that the storage modulus reduction rate was highest for the composites with the highest filler concentrations; this may arise due to the higher number of smaller-sized molecules incorporated into the polymer matrix [61,62].

Figure 7b shows the temperature dependence of the loss factor, tan δ, of the composites. Here, the T_g_ is given as the temperature corresponding to the maximum value of tan δ. The maximum tan δ for the TPU composites exhibited a shift toward higher temperatures corresponding with the addition of *h*-BN and OH-BN particles. This implies that the addition of fillers restricts the chain mobility and overall flexibility of the polymer molecules, thus increasing the T_g_ [58]. In addition, tan δ declined after the addition of filler because the presence of the *h*-BN and OH-BN particles reduced the damping capacity of the TPU matrix. The tan δ values for the OH-BN/TPU composites were lower than those of the *h*-BN/TPU composites. This is attributed to the increased interfacial adhesion between the OH-BN fillers which thereby reduce the energy dissipation in the composite material [63,64]. Consequently, these observations further confirm the positive effects of hydroxylated surface modification on the interfacial interaction between BN-type fillers and the TPU matrix.

### 3.6. Dielectric Constant

For applications in electrical device packaging, low dielectric constants, low dielectric loss and high resistivity are also required for TIMs to decrease the time delay of signal propagation and power dissipation. The BN-type TPU composites were therefore examined to determine their performance in this regard. Figure 8 shows the variation in the dielectric constant and dielectric loss tangent as a function of frequency for pure TPU and the various composites. As depicted in Figure 8a, the dielectric constant of pure TPU was extremely low, with a value of 3.26 at 1 kHz. Conversely, the composites possessed higher dielectric constants, which increased with the filler concentration; this is because the *h*-BN and OH-BN have intrinsically higher dielectric constants than TPU, and the increasing number of inorganic fillers therefore introduced more charge carriers into the composite matrix. The maximum dielectric constant of 3.97 was achieved for the OH-BN/TPU composite at a 10 wt.% filler concentration [65,66,67].

Figure 8b shows that the dielectric loss tangent of the composites decreases as the frequency is increased. This is because the interfacial polarization requires sufficient time and cannot keep pace with the external electric field at higher frequencies [68,69,70]. However, the dielectric constants and dielectric loss tangents of the OH-BN/TPU composites were higher than that of the *h*-BN/TPU composites at the same filler concentration; this can be attributed to the lone pair of electrons in the hydroxyl groups of the functionalized OH-BN materials, which increased the polarity of the composite material [67]. Overall, the dielectric constants and dielectric loss tangents of all the TPU composites were maintained, particularly at low frequencies, indicating that these materials possess satisfactory dielectric stabilities.

### 3.7. Thermal Conductivity

Figure 9 shows the thermal conductivities of the FDM 3D-printed *h*-BN/TPU and OH-BN/TPU composites with filler loading concentrations of 0, 5 and 10 wt.%. Considering the unique stacking process inherent in this printing technique, the thermal conductivities were compared between the different printing orientations (SV and FP). For all the composites, the thermal conductivity increased with increasing filler content because of the progressive introduction of more thermally conductive paths through the polymer matrix. Moreover, at the same filler loading, the thermal conductivity of the SV-orientated specimens was consistently higher than in the FP-oriented composites. It is well known that when the filler particles are orientated in the same direction as the heat flow, the thermal conductivity considerably improves. The higher thermal conductivity of the SV-printed samples can therefore be attributed to the BN-type fillers being orientated mainly along the filament printing direction, thus parallel to the heat flow direction. In contrast, the relatively lower thermal conductivity values of the FP-printed samples arises from the interfaces between the printing filaments, where the heat flow direction is perpendicular to the orientation of *h*-BNs in the composite filament. Continuous phonon transmission is therefore impeded, which further increases the interface thermal resistance and leads to poor heat conduction. Furthermore, with a 10 wt.% filler concentration, the thermal conductivity of the OH-BN/TPU (SV) composite reached 0.75 W∙m^−1^∙K^−1^, approximately 2.5 times greater than that of the comparable *h*-BN/TPU (SV) sample. This indicates that the hydroxyl groups in the OH-BN particles make a major contribution to increasing the thermal conductivity. As indicated in the earlier analyses, the OH-BN particles were dispersed homogenously throughout the TPU matrix, which enabled the formation of continuous heat conduction networks through the composite materials. In addition, hydrogen bonding between the hydroxyl groups and the TPU polymer chains improves the interface compatibility, thus reducing the air gap between the filler particles and the matrix and therefore minimizing the heat conduction resistance.

## 4. Conclusions

In this study, hydroxyl-functionalized OH-BN particles were prepared to explore their impact on the surface compatibility, mechanical properties and the final anisotropic thermal conductivity of thermoplastic polyurethane composites through simple FDM 3D-printing processes. The functionalization process is facile and reliable, producing ceramic-type filler particles that can be dispersed homogeneously through a polymer matrix, as confirmed by FE-SEM. The resulting chemical interactions between OH-BN and the TPU matrix were revealed by FT-IR, while Raman analysis confirmed that the reduced-layer structure of the OH-BN particles compared to unmodified *h*-BN fillers. The OH-BN/TPU composites displayed a notable increase in tensile strength and storage modulus, owing to the excellent dispersibility and strong adhesion properties of the OH-BN particles, which induced efficient load transfer at the filler/matrix interface. Furthermore, the OH-BN fillers display considerable potential for enhancing the thermal conductivities of TIM composites, particularly when the composite is printed in such a way that facilitates the parallel orientation of the filler particles along the filament structure. These materials possess excellent electrical insulation and dielectric properties which enable their future integration in the electronics industry and in new-energy automobiles. This study therefore provides new possibilities for the additive manufacturing of TIMs with excellent thermal conductivities and mechanical properties.

## Figures and Tables

**Figure 1 materials-16-00356-f001:**
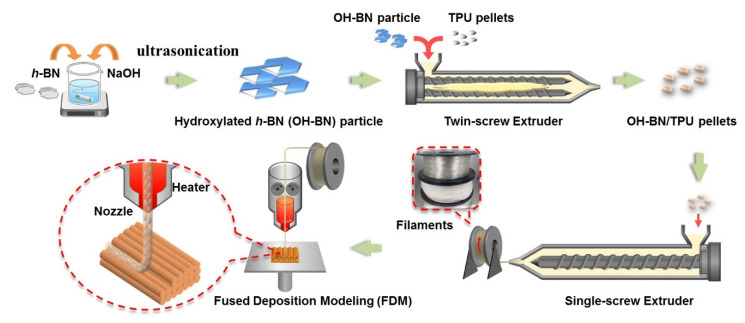
Schematic of the full preparation procedure of the OH-BN/TPU and h-BN/TPU filaments and their 3D-printed composites.

**Figure 2 materials-16-00356-f002:**
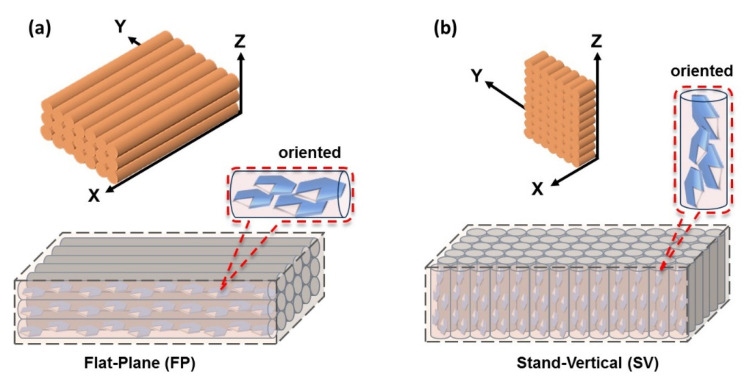
Schematics of the orientation of the OH-BN/*h*-BN particle networks in the composite filaments prepared by two printing methods: (**a**) flat-plane, and (**b**) stand-vertical.

**Figure 3 materials-16-00356-f003:**
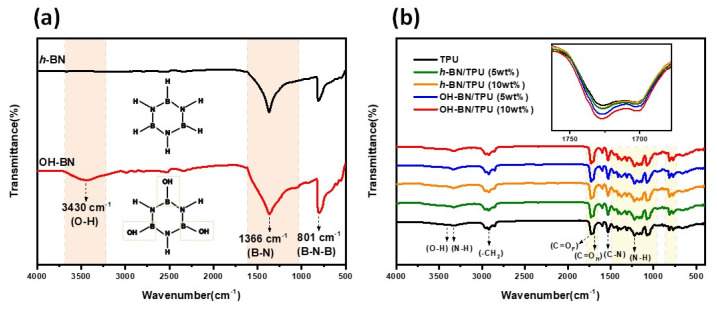
FT-IR spectra of (**a**) the *h*-BN and OH-BN particles; (**b**) pure TPU compared to the *h*-BN/TPU and OH-BN/TPU composites (at 5 and 10 wt.% filler concentrations).

**Figure 4 materials-16-00356-f004:**
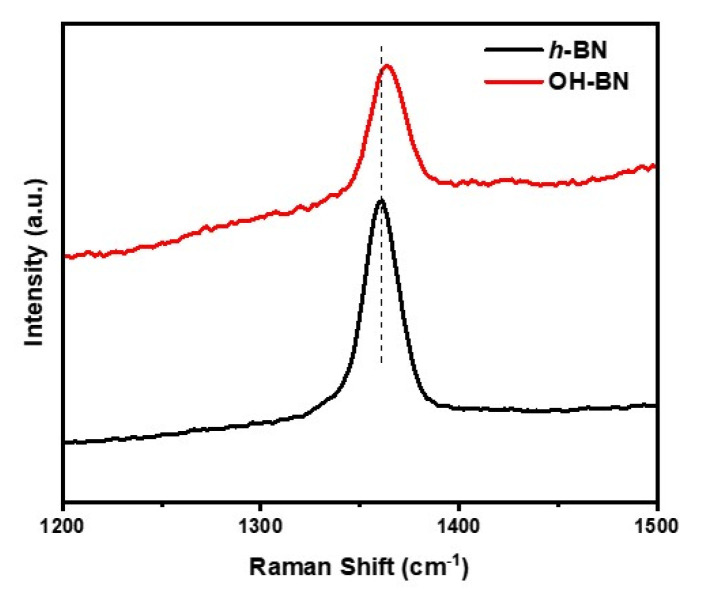
Raman spectra of the *h*-BN (black) and OH-BN (red) particles.

**Figure 5 materials-16-00356-f005:**
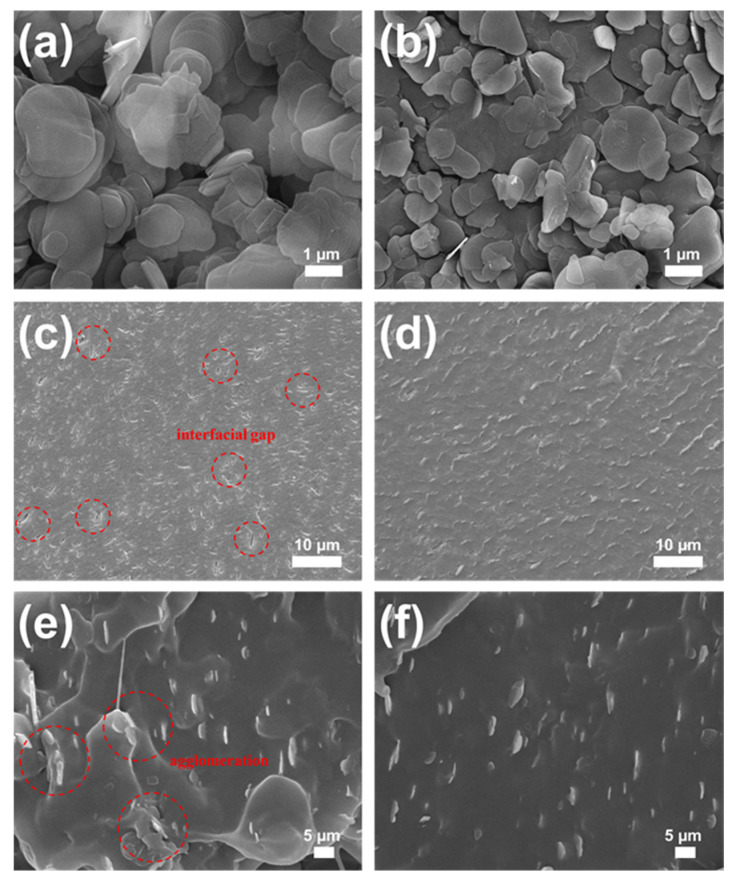
FE-SEM images of (**a**) *h*-BN and (**b**) OH-BN filler particles; (**c**) cross-sectional view of the *h*-BN/TPU and (**d**) OH-BN/TPU filaments; (**e**) the *h*-BN/TPU and (**f**) OH-BN/TPU composite cross-sectional morphologies. Each composite studied incorporated a 10 wt.% filler concentration.

**Figure 6 materials-16-00356-f006:**
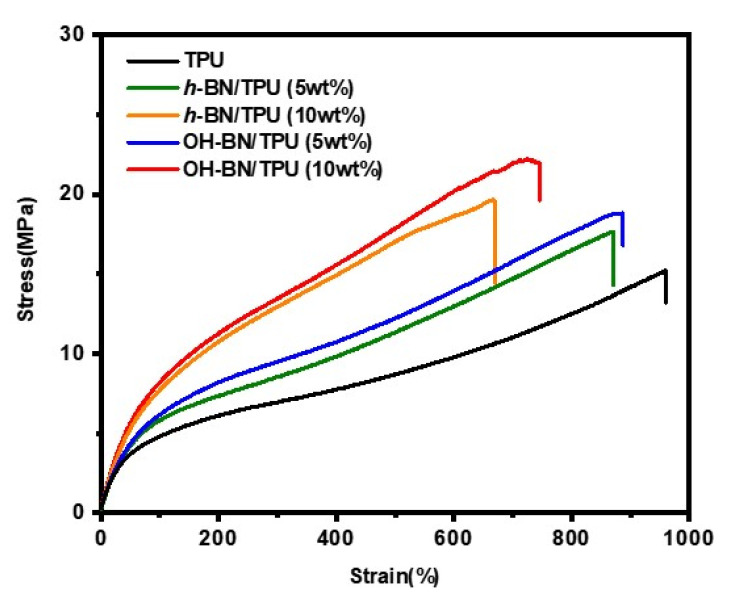
The stress–strain dependence of pure TPU compared with the *h*-BN/TPU and OH-BN/TPU composites (at 5 and 10 wt.% filler concentrations).

**Figure 7 materials-16-00356-f007:**
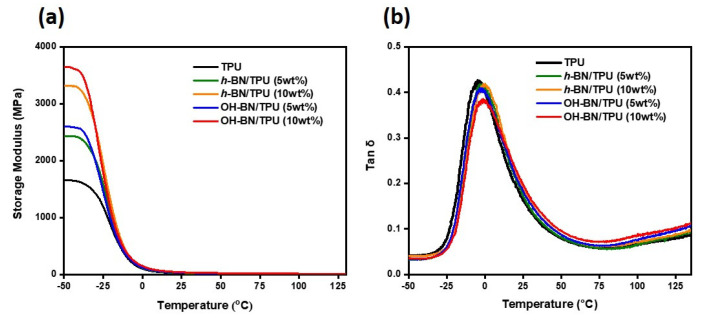
(**a**) Storage modulus and (**b**) loss factors (tan δ) versus temperature of pure TPU compared with the *h*-BN/TPU and OH-BN/TPU composites (at 5 and 10 wt.% filler concentrations).

**Figure 8 materials-16-00356-f008:**
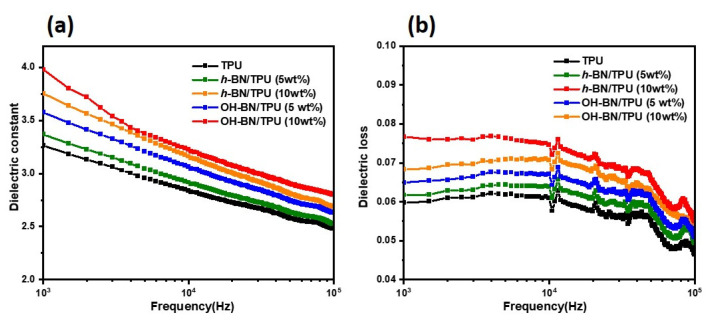
(**a**) Dielectric constants and (**b**) dielectric loss tangents as a function of frequency for pure TPU compared to the *h*-BN/TPU and OH-BN/TPU composites (at 5 and 10 wt.% filler concentrations).

**Figure 9 materials-16-00356-f009:**
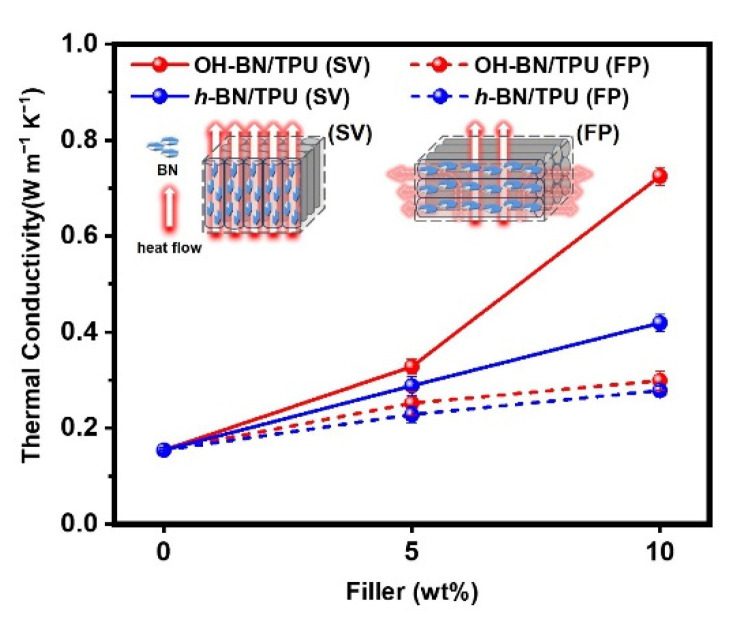
Thermal conductivities of the *h*-BN/TPU and OH-BN/TPU composites at various filler concentrations, printed in either an SV or FP orientation. The heat conduction pathways through the composites according to the printing orientation are shown in the figure inset.

## Data Availability

Original data can be requested to the corresponding author.

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
