# Peer review of "Improvement of the Thermal Conductivity and Mechanical Properties of 3D-Printed Polyurethane Composites by Incorporating Hydroxylated Boron Nitride Functional Fillers"

_materials, 2022, doi:10.3390/ma16010356_

Round 1

Reviewer 1 Report

The paper will be ready for publication after major revision according to the attached file.

Reviewer 2 Report

The study presented in this research is sound, and the results produced are interesting. But a revision is required, and after responding to the following remarks and revising the paper, the manuscript may be considered for publication.

1. Literature review needs to include several recent, relevant publications (high impact) highlighting their key findings. The current version only discussed general aspects while the review of each from several papers is necessary. You may provide a review summary table consisting of a column for the comments or key conclusions.

2. More recent relevant literature or similar work discussion is mandatory in the introduction section, which is missing in the Introduction. Authors are suggested to add one paragraph in the introduction section by discussing the recent progress and citing similar work.

3. The novelty of the work is missing in the introduction. Authors are suggested to include a separate paragraph discussing the novelty and importance of the present work.

4. Authors are suggested to include a literature review on the recent publication on composite materials based on the following references in the introduction section: DOIs: 10.1016/j.rinp.2018.06.010; 10.1088/2053-1591/ab22d8; 10.1016/j.rinp.2019.102264; 10.1080/10420150.2019.1606809.

5. Reduce the similarity. Check the attached similarity report.

6. Also, check the typos throughout the manuscript during revision submission.

Reviewer 3 Report

This is a timely effort by the authors on"Improvement of the Thermal Conductivity and Mechanical Properties of 3D-printed Polyurethane Composites by Incorporating Hydroxylated Boron Nitride Functional Fillers". And it is well written. However, there are few suggestions.

1. Novelty needs to be more highlighted with its application in materials world.

2. Couple of spelling mistakes are there throughout this manuscript which must be proofread.

3. Are all the testing methods and material preparation methods standardized?

4. In Figure 5, the 'b' sub-part of Figure is repeated. 

5. Why did you select a particular set of values of FDM factors for preparing 3D printed TPU composites in the presence of more options?

Round 2

Reviewer 1 Report

Accept.